# Stimulation of the Pro-Resolving Receptor Fpr2 Reverses Inflammatory Microglial Activity by Suppressing NFκB Activity

**DOI:** 10.3390/ijms242115996

**Published:** 2023-11-06

**Authors:** Edward S. Wickstead, Bradley T. Elliott, Sarah Pokorny, Christopher Biggs, Stephen J. Getting, Simon McArthur

**Affiliations:** 1Institute of Dentistry, Faculty of Medicine & Dentistry, Queen Mary University of London, Blizard Institute, 4, Newark Street, London E1 2AT, UK; 2School of Life Sciences, College of Liberal Arts & Sciences, University of Westminster, 115, New Cavendish Street, London W1W 6UW, UK; 3Icahn School of Medicine at Mount Sinai, Department of Neurology, Simon Hess Medical and Science Building, New York, NY 10029, USA

**Keywords:** neuroinflammation, oxidative stress, microglia, resolution

## Abstract

Neuroinflammation driven primarily by microglia directly contributes to neuronal death in many neurodegenerative diseases. Classical anti-inflammatory approaches aim to suppress pro-inflammatory mediator production, but exploitation of inflammatory resolution may also be of benefit. A key driver of peripheral inflammatory resolution, formyl peptide receptor 2 (Fpr2), is expressed by microglia, but its therapeutic potential in neurodegeneration remains unclear. Here, we studied whether targeting of Fpr2 could reverse inflammatory microglial activation induced by the potent bacterial inflammogen lipopolysaccharide (LPS). Exposure of murine primary or immortalised BV2 microglia to LPS triggered pro-inflammatory phenotypic change and activation of ROS production, effects significantly attenuated by subsequent treatment with the Fpr2 agonist C43. Mechanistic studies showed C43 to act through p38 MAPK phosphorylation and reduction of LPS-induced NFκB nuclear translocation via prevention of IκBα degradation. Here, we provide proof-of-concept data highlighting Fpr2 as a potential target for control of microglial pro-inflammatory activity, suggesting that it may be a promising therapeutic target for the treatment of neuroinflammatory disease.

## 1. Introduction

Neuroinflammation is a complex biological response required to protect the central nervous system (CNS) from injury or infection. Microglia, the resident macrophage-like innate immune cells of the CNS, are essential to this process [1,2], with shifts in cellular activation being crucial to eliminate the noxious insult, but also to facilitate inflammatory resolution and tissue repair [3]. However, if this homeostatic resolution response becomes impaired it can result in the eventual development of self-perpetuating, chronic neuroinflammation, adding to tissue damage and aggravating disease, as has been reported following trauma, ischaemic-reperfusion injury or during neurodegenerative disease [4,5,6].

In the absence of currently impractical population-wide screening, neuroinflammation is generally not identified until after disease diagnosis, and thus curative rather than preventative therapies are likely to be required for efficient clinical management. Activating inflammatory resolution, the endogenous process through which the acute inflammatory response is curtailed and tissue homeostasis restored [7], is a potentially attractive solution to this problem. Although most widely studied in the peripheral immune response, it is thought that similar pro-resolving circuitry underlies the microglial response in neuroinflammation [8]. Thus, actively stimulating inflammatory resolution rather than simply inhibiting pro-inflammatory mediators could hold promise as a new treatment approach for the neuroinflammatory pathology associated with many neurodegenerative diseases. 

The formyl peptide receptors (FPRs) are G-protein coupled receptors with roles in chemotaxis, host defence and inflammation [9]. Of these, FPR2 or its murine functional homologues, Fpr2 and Fpr3, has a key role in peripheral inflammatory resolution [10,11,12], regulating monocyte/macrophage recruitment [9,10,13], phenotypic expression and behaviour [14,15]. Important protective actions of these receptors have been identified in diverse inflammatory settings in both mice and humans, including for example viral infection [16], mucosal injury [17] and acute heart failure [18]. Notably, FPR2 is a highly promiscuous receptor with a wide variety of small molecule, lipid, peptide and protein ligands that can exert opposing effects on inflammatory processes, e.g., β-amyloid peptide, LL-37 and serum amyloid A are all known pro-inflammatory FPR2 ligands, while lipoxin A_4_ and annexin A1 (ANXA1) are potent anti-inflammatory agonists [19], effects thought to be due to alternative hetero/homodimerization and intracellular signalling pathway activation [20]. 

Expression of FPR2 has been described for several different brain cell types, but the strongest evidence supports its presence in microglia where its expression is significantly up-regulated by pro-inflammatory stimuli [21,22,23]. Several studies have shown activation of FPR2 with anti-inflammatory agonists to lessen the impact of subsequent neuroinflammatory challenge, reducing pro-inflammatory cytokine and reactive oxygen species production [23,24,25] and stimulating an anti-inflammatory microglial phenotype [26], suggesting activation of the receptor may be able to prevent neuroinflammatory activity. It is less clear however, whether activation of FPR2 in the brain could affect pre-existing neuroinflammation, i.e., what potential FPR2 activation might have to accelerate inflammatory resolution. We therefore sought to investigate the relationship between the receptor and microglial behaviour, focusing on whether anti-inflammatory FPR2 agonist treatment could induce cells activated by bacterial lipopolysaccharide (LPS) exposure to revert to a more homeostatic phenotype in vitro. 

## 2. Results

### 2.1. Fpr2 Stimulation Reverses the Pro-Inflammatory Effects of LPS upon Microglia

We used the BV2 immortalised microglial cell line as an experimental model, a cell type we have previously shown to express the major murine FPRs Fpr1 and Fpr2 [27], exposed to the potent pro-inflammatory stimulus LPS at a concentration (50 ng/mL) we have previously shown to induce inflammatory activation [27]. Administration of the Fpr2/3 agonist C43 (100 nM) 1 h after LPS treatment (50 ng/mL) reversed LPS-induced production of the pro-inflammatory mediators nitric oxide (measured as soluble nitrite) and TNFα 48 h later (Figure 1A,B), although without noticeably affecting the LPS-stimulated induction of inducible nitric oxide synthase (iNOS) expression (Appendix A). In contrast, LPS-induced production of the anti-inflammatory cytokine IL-10 was significantly enhanced at 48 h by subsequent C43 treatment (Figure 1C). 

A negative aspect of neuroinflammation is the instigation of damage by microglia to otherwise healthy bystander cells and their subsequent removal by phagocytosis [28], we therefore investigated the ability of Fpr2/3 stimulation to ameliorate these effects. While treatment with C43 alone (100 nM) did not affect BV2 cell phagocytosis of healthy CMFDA-labelled PC12 cells, treatment with LPS (50 ng/mL; 24 h) substantially increased the proportion of BV2 cells phagocytosing PC12 cells. This was no longer apparent when LPS-stimulated cells were further treated with C43 (100 nM, 1 h post-LPS; Figure 1D), further indicating that that activation of Fpr2/3 can stimulate processes aiming to terminate inflammation.

The role of metabolic pathways in governing immune cell function has steadily become apparent over recent years, with inflammatory cells tending to favour glycolysis over oxidative phosphorylation for energy generation [29]. Having first confirmed that neither C43 nor LPS treatment affected cell proliferation (Appendix A), we therefore investigated whether Fpr2/3 activation could modulate the effects of LPS stimulation upon glucose utilisation and L-lactate production, indirect measures of glycolysis [30]. LPS (50 ng/mL) significantly increased L-lactate production from BV2 microglia at 48 h, an effect reversed by C43 administration 1 h post-LPS treatment (Figure 1E). Similarly, while LPS (50 ng/mL) reduced BV2 culture medium glucose content 48 h post stimulation, the medium glucose content in cultures treated with LPS and then subsequently with C43 (100 nM, 1 h post-LPS) was not significantly different from control conditions (Figure 1F).

To confirm that the effects of C43 upon inflammatory parameters were mediated through its primary receptor Fpr2/3, we examined how inclusion of the Fpr2/3-selective antagonist WRW_4_ affected the ability of C43 to regulate inflammatory mediator production. Notably, the inhibitory effects of C43 upon LPS-stimulated nitric oxide and TNFα production, and the stimulatory effect of C43 upon IL-10 production, were all attenuated by 10 min pre-treatment with WRW_4_ at 10 µM (Figure 2A–C), strongly indicating a mediatory role for this receptor. Moreover, although C43 is a potent Fpr2 agonist, there is evidence that it can also stimulate the related human receptor FPR1, raising the possibility that the same might be true for its murine equivalent, Fpr1 [31]. We therefore verified whether the anti-inflammatory effects of C43 treatment were sensitive to the specific Fpr1 antagonist cyclosporin H; pre-treatment with this antagonist (0.7 µM, 10 min prior to C43) had no effect on the suppressive effects of C43 on either LPS-stimulated nitric oxide or TNFα release (Figure 2D,E), suggesting that this receptor is not involved in C43 activity. As endogenously produced ANXA1 is a critical pro-resolving mediator for numerous immune cell types and is expressed by microglia [27,32], we examined whether the effects of C43 were mediated through this protein. However, no changes in ANXA1 expression were identified, and shRNA for ANXA1 had no effect on the cytokine changes elicited by C43 (Appendix A). Together, these data strongly indicate C43 as acting through stimulation of Fpr2/3. 

To determine whether Fpr2/3 stimulation by C43 would still alter microglial phenotype after a longer interval, we examined expression of different phenotypic markers, namely the pro-inflammatory (CD38, CD40, CD86) and pro-resolving (CD206) cell surface proteins. Although CD86 did not respond to LPS stimulation, expression of both CD38 and CD40 were upregulated by LPS (48 h, 50 ng/mL), an effect reversed when cells were subsequently treated with C43 (100 nM, 24 h post-LPS; Figure 3A,B). In contrast, surface expression of the classically anti-inflammatory, pro-resolution marker CD206 was suppressed following LPS treatment, but this was rescued by subsequent C43 treatment (Figure 3C). C43 had no effect on any of these markers when administered alone.

BV2 cells are a useful and widely employed experimental microglial model, but as an immortalised cell line they may not always reflect the behaviour of true microglia. We therefore examined whether Fpr2/3 signalling could also control the inflammatory response to LPS in primary murine microglia. Treatment of primary microglia with LPS (50 ng/mL, 24 h) significantly increased the ratio of pro-inflammatory CD45^+ve^, F4/80^+ve^, Ly6C/G^+ve^ cells to anti-inflammatory CD45^+ve^, F4/80^+ve^, Ly6C/G^−ve^ cells [15], an effect reversed by subsequent C43 treatment (100 nM, 1 h post-LPS), notably this was prevented by pre-treatment with the Fpr2/3 antagonist WRW4 at 10 µM (Figure 4A,B). Analysis of medium L-lactate and glucose content indicated a significant increase in L-lactate production upon LPS stimulation, but no other effects reached statistical significance (Figure 4C,D). Taken together, these analyses of BV2 and primary microglia strongly indicate that selective Fpr2/3 stimulation with C43 can significantly diminish LPS-elicited microglial activation. 

### 2.2. LPS Induces ROS Production through Both the Mitochondria and NADPH Oxidase Activation, a Response Reversed by Fpr2/3 Agonist C43

A key anti-bacterial property of activated immune cells is the production of reactive oxygen species (ROS) in response to stimulation by bacterial components such as LPS [33]. However, uncontrolled ROS production can be highly damaging to bystander cells, an effect that in the brain can be a significant source of neuroinflammatory damage [34]. Strategies to reduce this activity may thus be of great benefit in the treatment of neuroinflammation. To determine whether Fpr2/3 activation could influence LPS-induced ROS production, BV2 microglia in culture were stimulated with either saline or LPS (50 ng/mL) for 10 min, then C43 was added (100 nM) and cellular ROS production was monitored every 5 min for 1 h. LPS increased ROS production by approximately 100% compared to untreated cells, production that was completely reversed by C43 treatment (Figure 5A,B). Approximately 30% of the observed increase in ROS following LPS stimulation could be accounted for by mitochondrial ROS production, with this also being reversed by C43 administration (Figure 5C,D).

Aside from mitochondrial production, the main source of microglial ROS in response to inflammatory stimulation is through activation of NADPH oxidase, also termed NOX2 [35]. NOX2 is a multi-subunit enzyme, with its activation requiring translocation of the p67phox subunit from the cytosol to the plasma membrane-bound gp91phox subunit [36]. 

Confocal microscopic analysis of BV2 cells stimulated with LPS for 30 min (50 ng/mL) indicated clear co-localisation of p67phox and gp91phox subunits at the plasma membrane, an effect that was prevented by treatment with C43 (100 nM) 10 min post-LPS (Figure 5E). These Fpr2/3 mediated effects were independent of the antioxidant systems involving heme oxygenase-1 (HO-1) and superoxide dismutase-2 (SOD2), with no changes in their expression (Appendix A). Thus, selective Fpr2/3 stimulation can successfully reverse LPS-induced mitochondrial and NADPH oxidase associated ROS production, suggesting upstream signalling pathway interference. 

### 2.3. C43 Signals through p38 MAPK but Not ERK1/2

Having established that Fpr2/3 stimulation can at least partially reverse pro-inflammatory microglial activity, we sought to identify the primary signalling pathway triggered by C43 in BV2 cells. We have previously shown that the Fpr2 agonist ANXA1 can upregulate IL-10 production via a p38 MAP kinase dependent mechanism [20], hence we examined this pathway in BV2 cells using a phosphorylated p38 MAP kinase ELISA. Treatment of BV2 cells with C43 (100 nM) rapidly increased p38 MAP kinase phosphorylation, reaching statistical significance at 2 min post-treatment (Figure 6A,B). In contrast, while Fpr2/3 can also signal through ERK1/2 MAP kinase [20], we did not see any sign that C43 could activate this signalling pathway over the same time period (Appendix A). To test whether activation of p38 MAP kinase signalling was relevant to the reversal of LPS effects, we examined the impact of the selective p38 MAP kinase inhibitor SB203580 upon nitrite accumulation. Notably, pre-treatment with SB203580 (2 µM, 10 min) prevented C43 (100 nM, 1 h post-LPS) from inhibiting LPS-induced (50 ng/mL, 24 h) nitrite production (Figure 6C), validating a role for p38 MAPK in mediating the anti-inflammatory effects of C43. 

### 2.4. C43 Reduces NFκB Activation by Decreasing LPS-Induced IκBα Degradation 

Classically, stimulation of NO and TNFα production by LPS is mediated through nuclear translocation of NFκB and subsequent transcriptional activity [37]; we therefore examined whether this pathway was involved in the response of BV2 cells to LPS, and whether it could be modulated by Fpr2/3 stimulation. Confocal microscopic analysis of NFκB p65 subunit localisation within BV2 cells confirmed that 30 min LPS exposure (50 ng/mL) induced its nuclear translocation, an effect attenuated by subsequent stimulation with C43 (100 nM, 10 min post-LPS; Figure 7A). This effect of C43 was inhibited by pre-treatment with the p38 MAP kinase inhibitor SB203580 (2 µM, 5 min prior to C43; Figure 7A). NFκB p65 is held in the cytoplasm by the regulatory IκBα complex [38], proteolytic degradation of which can be stimulated by LPS-induced phosphorylation [39]. Analysis confirmed that LPS treatment (30 min, 50 ng/mL) reduced total BV2 cell IκBα expression, an effect prevented by subsequent treatment with C43 (100 nM, 10 min post-LPS; Figure 7B,C). Together these data suggest that the anti-inflammatory effects of C43 are mediated through p38 MAPK activation and subsequent inhibition of NFκB nuclear translocation. 

## 3. Discussion

Alongside their central roles in neural circuit development, synaptic pruning and plasticity [40,41,42], microglia are critical for tissue maintenance, the response to injury [6] and defence against invading pathogens [43]. However, hyperactivation of microglia can become pathological, leading to chronic neuroinflammation and neurodegeneration [44]. Evidence for the role of neuroinflammation in brain disease is widespread [45], with clinical reports emphasising the significance of neuroinflammation in neurodegenerative conditions such as stroke [46], multiple sclerosis [47], and Alzheimer’s disease [48]. This is also underlined through improved disease outcomes following the application of anti-inflammatory and immunomodulatory therapeutic approaches [49,50]. Due to the potential for microglial activation to become damaging, strategies to control excessive inflammatory actions through promotion of a pro-resolving phenotype may be of significant benefit for a wide range of neurological diseases. In this study, we have used an in vitro cellular model to provide proof-of-principle evidence that targeting the pro-resolving receptor Fpr2/3 is a viable approach to restraining inflammatory microglial behaviour.

Murine Fpr2/3 and human FPR2 have central roles in the resolution of peripheral inflammation [9,10,15]. Despite also being expressed in microglia [24,27], relatively little is known about the receptor’s potential as an anti-neuroinflammatory target. Here we provide further evidence for the pro-resolving effects of Fpr2/3 agonists and extend these to CNS immune cell populations. Notably, our finding that post-LPS treatment with an Fpr2/3 specific agonist can significantly attenuate and reverse induction of a pro-inflammatory phenotype in BV2 microglia suggests that this receptor may have potential for exploitation as a therapeutic, rather than a prophylactic target. In particular, we suggest that activation of Fpr2/3 has potential to protect tissue from ongoing neuroinflammatory damage and is therefore a valuable target for further investigation of in vivo models of neuroinflammatory disease.

Our data moreover, indicate an important role for Fpr2/3 in restraining NFκB signalling via inhibition of IκBα degradation, extending our understanding of the receptor’s intracellular signalling pathways [20], as well as highlighting its potential to modulate pro-inflammatory signalling. The modulation of LPS-induced NFκB by Fpr2/3 stimulation has been reported recently in both monocyte and osteoclast [51] cell lines, suggesting it may be a common feature of macrophage-lineage cells. As microglia are known to rapidly upregulate Fpr2/3 expression following inflammatory insult [52], this serves as a further indication for the importance of this receptor in inflammatory regulation, emphasising the concept that inflammatory resolution is an intrinsic, programmed part of the inflammatory response [53].

In addition, we report for the first time that Fpr2/3 activation can reverse LPS-induced ROS production from two independent sources: the mitochondria and NADPH oxidase. Previous studies report the ability of LPS to induce ROS production in microglia via these pathways [54], both of which may be crucial in shifting microglial metabolic parameters [55] and triggering neuroinflammation [56]. This may be central to microglia associated neurodegeneration [57,58]. Moreover, in addition to their clear role in neurodegeneration [44], microglia may well become pathological during other chronic inflammatory conditions. For example, microglia have been shown to be crucial in the induction of chronic pain in mice in the absence of nerve injury [59]. Thus, modulating microglia phenotype may hold additional premise in disorders wherein neurons do not become inherently damaged.

Our work is not without its limitations, most notable of which is that our findings derive from the study of immortalised or primary microglia in mono-culture, a condition which essentially ignores the significant contributions to neuroinflammatory processes made by other brain cell types [60]. Similarly, our data identify some differences in behaviour between immortalised BV2 cells and primary microglia, e.g., in terms of the effects of C43 upon L-lactate production, potentially due to the significantly greater degree of between-culture variation seen in primary vs. immortalised microglia. These discrepancies together highlight the need for future work to determine whether the in vitro findings reported here can be translated into in vivo models. However, combined with the previous reports that pre-treatment with Fpr2/3 agonists guards against neuroinflammatory damage [23,24,25], our finding that Fpr2/3 stimulation can reverse inflammatory microglial behaviours strongly suggests that Fpr2/3 agonists capable of dampening pathological microglial activity could hold therapeutic promise for numerous neurological diseases.

## 4. Materials and Methods

### 4.1. Drugs and Reagents

The FPR2 agonist Compound-43 (*N*-(4-Chlorophenyl)-*N*-[2,3-dihydro-1-methyl-5-(1-methylethyl)-3-oxo-2-phenyl-1*H*-pyrazol-4-yl]-urea) and antagonist WRW_4_ (Trp-Arg-Trp-Trp-Trp-NH_2_) were purchased from Tocris Ltd., Bristol, UK. Isolated and purified lipopolysaccharides from *Escherichia coli*, serotype O111:B4 were purchased from Merck Millipore Ltd., Gillingham, UK.

### 4.2. Cell Culture

The BV2 murine microglial line and shRNA annexin A1 clones were kind gifts from Prof E. Blasi (Università degli Studi di Modena e Reggio Emilia, Modena, Italy) and Dr E. Solito (The William Harvey Research Institute, Queen Mary University of London, London, UK), respectively. Cells were cultured in DMEM supplemented with 5% heat-inactivated fetal calf serum (FCS) with 100 μM non-essential amino acids, 2 mM L-alanyl-L-glutamine and 50 mg/mL penicillin-streptomycin (Thermofisher Scientific, Horsham, UK) at 37 °C with 5% CO_2_ and 95% air. Prior to experimentation, BV2 cells were serum starved for 24 h. 

### 4.3. Primary Microglial Culture

Primary murine microglia were prepared from 8-week old male C57Bl/6 mice by papain dispersal and Percoll density gradient separation as described previously [32]. Cells were cultured in DMEM medium supplemented with 20% fetal calf serum and 100 μM non-essential amino acids, 2 mM L-alanyl-L-glutamine, and 50 mg/mL penicillin-streptomycin (all Thermofisher Scientific, UK) at 37 °C in 5% CO_2_, 95% air.

### 4.4. PrestoBlue Cell Proliferation Assay

Cell proliferation rate was determined using the PrestoBlue reagent (ThermoFisher Scientific, Loughborough, UK). PrestoBlue solution was diluted 1:10 in PBS prior to cellular incubation at 37 °C in the dark for 15 min. Fluorescence was measured using a CLARIOstar microplate reader (BMG Labtech, Ortenberg, Germany), with excitation and emission filters set at 560 and 590 nm, respectively. Confirmation that neither C43 nor LPS directly contributed to the fluorescent signal was determined in the absence of cells. Background autofluorescence was determined using unlabelled cells in PBS alone.

### 4.5. Griess Assay

Nitric oxide production was indirectly detected through the measurement of nitrite using the Griess method. Cellular supernatant was treated with Griess reagent (3.85 μM napthylethylenediamine dihydrochloride, 58.1 μM sulphanilamide, 5% ortho-phosphoric acid) at room temperature in the dark for 15 min. Absorbance was detected at 540 nm with a CLARIOstar microplate reader (BMG Labtech, Germany). Final concentrations were determined by reference to a standard curve of sodium nitrite in cell culture medium (3–100 μM).

### 4.6. Cytokine ELISA

Supernatant content of tumour necrosis factor alpha (TNFα) and interleukin-10 (IL-10) were assayed by murine-specific sandwich ELISAs using commercially available kits, according to the manufacturer’s instructions (Thermofisher Scientific, Loughborough, UK). A CLARIOstar spectrophotometer (BMG Labtech, Germany) was used to measure absorbance at 450 nm. Absorbance values recorded at 570 nm were subtracted to reduce optical interference. 

### 4.7. p38 MAPK ELISA

BV2 cells were treated according to experimental design. Samples were lysed and protein normalised following content assessment using Bradford reagent [61] for use in an InstantOne^TM^ total/phosphor multispecies p38 ELISA kit (Thermofisher Scientific, Loughborough, UK) according to the manufacturer’s instructions. Absorbance was measured at 450 nm and 570 nm, as previously described. 

### 4.8. Phenotypic Marker Expression

Following incubation with blocking buffer (0.01 M PBS,1 μg/mL anti rat anti-mouse CD16/32 monoclonal antibody, 1% FCS, 1 mM CaCl_2_) on ice for 15 min, BV2 cells were labelled with APC-conjugated rat anti-mouse CD38 (1:80) and CD40 (1:20), and PE-conjugated rat anti-mouse CD206 (1:40), and FITC-conjugated rat anti-mouse CD86 (1:80; Biolegend, London, UK) for 30 min on ice for analysis by flow cytometry. Immunofluorescence was analysed for 10,000 singlet events per sample using a BD FACSCanto II (BD Biosciences, Becton, UK) flow cytometer; data were analysed using FlowJo 8.8.1 software (Treestar Inc., Ashland, OR, USA).

### 4.9. PC12 Cell Phagocytosis

PC12 cells were labelled by incubation with 5 μM 5-chloromethylfluorescein diacetate (CMFDA; Thermofisher Scientific, Loughborough, UK) in serum free DMEM medium for 30 min at 37 °C, washed in serum free DMEM medium, and cultured with BV2 cells (previously treated according to experimental design) at a ratio of 3:1 for 1 h in a humidified incubator in 5% CO_2_ at 37 °C. Cells were extensively washed in cold PBS to separate non-engulfed cells, collected and cellular fluorescence was determined using a FACSCanto II flow cytometer (BD Biosciences, UK) equipped with a 488 nm laser and FlowJo 8.8.1 software (Treestar Inc., Ashland, OR, USA). A total of 10,000 single events were quantified per sample. 

### 4.10. Lactate & Glucose Determination

L-lactate production and glucose usage were simultaneously determined using a YSI 2300 Stat Plus machine (YSI Life Sciences Inc., Yellow Springs, OH, USA). Following experimental treatments, cell supernatant was collected and stored at −80 °C until required. Concentration readings for L-lactate and glucose were linear up to 30 mM/L and 50 mM/L, respectively.

### 4.11. Reactive Oxygen Species (ROS) Assays

Total intracellular ROS production was quantified using 6-chloromethyl-2′,7′-dichlorodihydrofluorescein diacetate, acetyl ester (CM-H_2_DCFDA; Thermofisher Scientific, UK) according to the manufacturer’s instructions. Cells were plated at 200,000 cells/cm^2^ in phenol red-free (PRF) DMEM and serum starved overnight prior to being pre-loaded with 5 µM CM-H_2_DCFDA for 15 min at 37 °C. Unbound dye was then removed, and fresh PRF-DMEM was added prior to experimentation. Following treatment administration, cellular fluorescence was determined every 5 min for 1 h at 37 °C using a CLARIOstar fluorescence microplate reader (BMG Labtech, Germany) with excitation and emission filters set to 492 nm and 517 nm, respectively. 

Mitochondrial superoxide production (mtROS) was quantified using the MitoSOX Red tracer (Thermofisher Scientific, UK) according to the manufacturer’s instructions, with a loading concentration of 2.5 μM. Following treatment, cellular fluorescence was determined every 5 min for 1 h at 37 °C using a CLARIOstar fluorescence microplate reader (BMG Labtech, Germany) with excitation and emission filters set at 510 nm and 580 nm, respectively.

### 4.12. Western Blot Analysis

Treated cells were collected into RIPA buffer (10 mM Tris-HCl, 1 mM EDTA, 0.5 mM EGTA, 1% Triton X-100, 0.1% sodium deoxycholate, 0.1% sodium dodecyl sulfate, 140 mM NaCl, pH 8.0; all Merck Millipore Ltd., Gillingham, UK) supplemented with protease and phosphatase inhibitor cocktails (Abcam Ltd., Cambridge, UK) and homogenised by repeated freeze-thaw cycles prior to protein content estimation. Samples boiled in 6× Laemmli buffer were subjected to standard SDS-PAGE (10%) prior to being electrophoretically blotted onto Immobilon-P polyvinylidene difluoride membranes (Merck, Gillingham, UK). Ponceau S staining (Merck, Gillingham, UK) was used to quantify total protein. Membranes were blotted using antibodies raised against murine superoxide dismutase 2 (SOD; rabbit monoclonal, 1:1000), haem oxygenase-1 (HO-1; rabbit polyclonal, 1:1000), IκBα (mouse monoclonal, 1:1000; Cell Signalling Technology Inc., Danvers, MA, USA) or NFκB p65 (1:400, clone D14E12, Cell Signaling Technologies Inc., USA) in Tris-buffer saline solution containing 0.1% Tween-20 and 5% (*w*/*v*) non-fat dry milk overnight at 4 °C. Membranes were subsequentially washed with Tris-buffer saline solution containing 0.1% Tween-20, prior to incubation with appropriate secondary antibody (horseradish peroxidase-conjugated goat anti-rabbit, 1:5000, Thermofisher Scientific, UK), for 90 min at room temperature. Proteins were then detected using enhanced chemiluminescence detection (0.4 mM p-coumaric acid, 7.56 mM H_2_O_2_ and 2.5 mM luminol in 1 M Tris, pH 8.5) and visualised on X-ray film (Scientific Laboratory Supplies Ltd., Nottingham, UK).

### 4.13. Immunofluorescence & Confocal Microscopy

Following experimental treatments, BV2 microglia in chambered culture slides were fixed by incubation with 2% formaldehyde in PBS for 10 min at 4 °C, washed and non-specific antibody binding was minimised through 30 min incubation at room temperature in PBS containing 10% hiFCS and 0.05% Triton-X 100 (Thermofisher Scientific, UK). Cells were then incubated with rabbit anti-mouse p67phox monoclonal antibody (1:500, clone EPR5064; Abcam Ltd., UK) and mouse anti-mouse gp91phox monoclonal antibody (1:50, clone 53, BD Biosciences, UK) or rabbit anti-mouse NFκB monoclonal antibody (1:400, clone D14E12, Cell Signalling Technology Inc., USA) overnight at 4 °C in PBS with 1% hiFCS and 0.05% Triton-X 100. Cells were washed and incubated with AF488-conjugated goat anti-mouse and AF647-conjugated goat anti-rabbit secondary antibodies (both 1:500, Thermofisher Scientific, UK) in PBS with 1% hiFCS and 0.05% Triton-X 100 at room temperature in the dark for 1 h. Cells were washed with PBS, nuclei stained with 180 nM DAPI in ddH_2_O for 5 min and mounted with Mowiol solution. Cells were imaged using an LSM710 confocal microscope (Leica Microsystems Ltd., Solihull, UK) fitted with 405 nm, 488 nm, and 647 nm lasers and a 63× oil immersion objective lens. Images were captured with ZEN Black 2.0 software (Zeiss Microscopy Ltd., Cambridge, UK) prior to analysis with ImageJ 1.51w (National Institutes of Health, Bethesda, MD, USA). Co-localisation of p67phox and gp91phox was determined using graded lookup tables for red and green pixels only. 

### 4.14. Statistical Analysis

Sample sizes were calculated to detect differences of 15% or more, with a power of 0.85 and α set at 5%; calculations being informed by previously published data [27]. All experimental data are presented as mean ± s.e.m., with a minimum of *n* = 3 independent cultures. All assays were performed in triplicate. For all data, the normality of distribution was established with the Shapiro-Wilk test, followed by further analysis with two-tailed Student’s *t* tests to compare two groups or, for multiple comparison analysis, one-, two-, or three-way ANOVA followed by Tukey’s HSD *post hoc* test. A *p* value of <0.05 was considered statistically significant. All statistical analyses were performed with Graph Pad Prism 8 software (GraphPad Software, San Diego, CA, USA).

## 5. Conclusions

This study has identified that the Fpr2 agonist C43 is a potent pro-resolving ligand, successfully modulating microglial phenotype and inflammatory cytokine release alongside ablating mitochondrial and NADPH oxidase induced ROS production, following an LPS-induced inflammatory response. This data therefore suggests that the modulation of Fpr2 could hold potential in the development of therapeutics for neuroinflammatory disease.

## Figures and Tables

**Figure 1 ijms-24-15996-f001:**
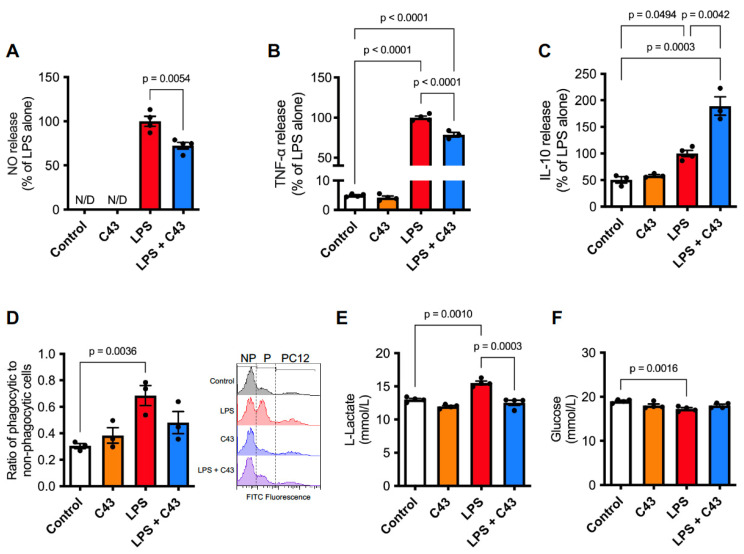
**Treatment with C43 reverses an LPS-induced pro-inflammatory phenotype in BV2 microglia.** (**A**,**B**) Treatment with C43 (100 nM; 1 h post-LPS) attenuates LPS-induced (50 ng/mL, 48 h) release of the pro-inflammatory mediators nitric oxide and TNFα. (**C**) Release of IL-10 from BV2 cells is stimulated by LPS (50 ng/mL, 48 h), an effect substantially enhanced by C43 treatment (100 nM, 1 h post-LPS). (**D**) LPS treatment (50 ng/mL, 24 h) stimulates phagocytosis of fluorescently labelled PC12 cells, an effect attenuated by C43 (100 nM, 1 h post-LPS), presented alongside typical flow cytometry histograms (NP: non-phagocytic, P: phagocytic, PC12: non-apoptotic PC12 cells). (**E**) LPS treatment (50 ng/mL, 48 h) increases BV2 cell medium L-lactate content, an effect attenuated by C43 treatment (100 nM, 1 h post-LPS). (**F**) LPS treatment (50 ng/mL, 48 h) reduced medium glucose content, this effect was no longer significant following C43 treatment (100 nM, 1 h post-LPS). Data are mean ± s.e.m. of 3–4 independent cultures.

**Figure 2 ijms-24-15996-f002:**
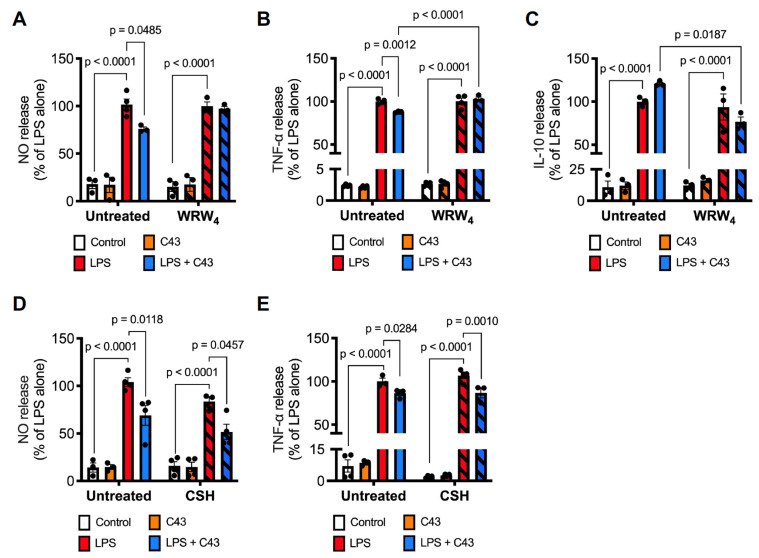
**The effects of C43 on microglial inflammatory mediator production are sensitive to the Fpr2/3 antagonist WRW_4_ but not the Fpr1 antagonist cyclosporin H.** (**A**,**B**) Pre-treatment with WRW_4_ (10 μM, 10 min prior to C43) blocked the reversal of LPS (50 ng/mL; 48 h) induced nitric oxide (**A**) or TNFα (**B**) production caused by C43 (100 nM, 1 h post-LPS). (**C**) Pre-treatment with WRW_4_ (10 μM, 10 min prior to C43) prevented C43 (100 nM, 1 h post-LPS) induced enhancement of LPS (50 ng/mL, 48 h)-elicited IL-10 production. (**D**,**E**) Pre-treatment with CSH (0.7 μM, 10 min prior to C43) had no effect on the reversal of LPS (50 ng/mL; 48 h) induced nitric oxide (**D**) or TNFα (**E**) production caused by C43 (100 nM, 1 h post-LPS). Data are mean ± s.e.m. of 3–4 independent cultures.

**Figure 3 ijms-24-15996-f003:**
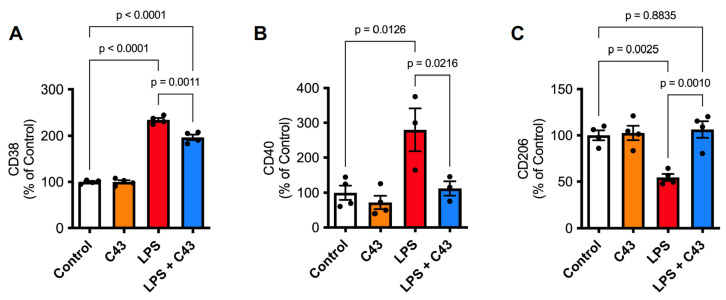
**Delayed treatment with C43 reverses LPS-induced changes in cell surface phenotypic markers in BV2 microglia.** (**A**,**B**) Treatment with C43 (100 nM; 24 h post-LPS) attenuated the increase in cell surface (**A**) CD38 and (**B**) CD40 expression induced by LPS treatment (50 ng/mL, 48 h). (**C**) Treatment with C43 (100 nM; 24 h post-LPS) attenuated the reduction in cell surface CD206 expression induced by LPS treatment (50 ng/mL, 48 h). Data are mean ± s.e.m. of 3–4 independent cultures.

**Figure 4 ijms-24-15996-f004:**
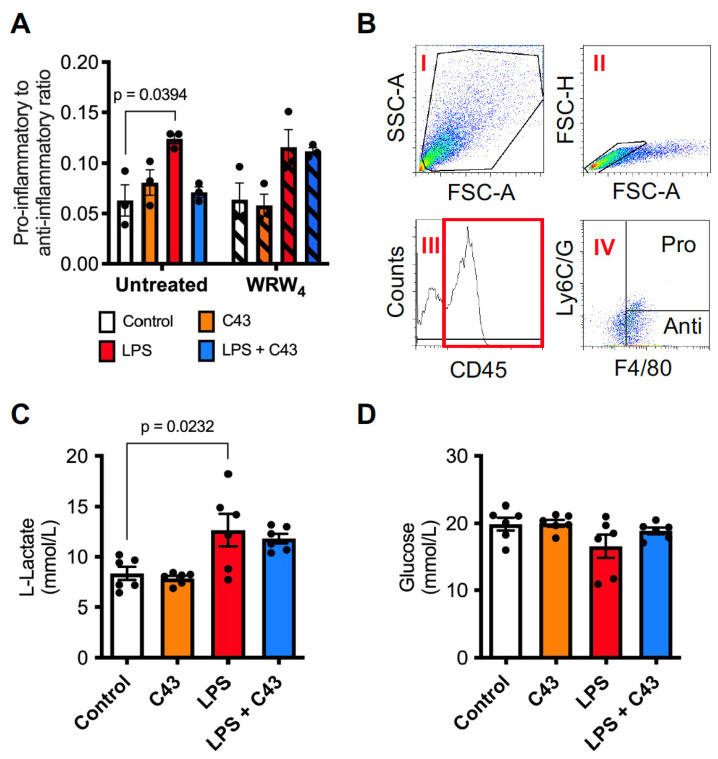
**Treatment with C43 reverses LPS-induced pro-inflammatory phenotypic changes in primary microglia through Fpr2/3.** (**A**) Treatment with C43 (100 nM; 1 h post-LPS) attenuated the LPS-induced (50 ng/mL, 24 h) increase in pro-inflammatory to anti-inflammatory primary murine microglia ratio, an effect prevented by WRW_4_ pre-treatment (10 µM, 10 min prior to LPS). (**B**) Gating strategy depicting identification of pro- (CD45+, F4/80+, Ly6C/G+) and anti-inflammatory (CD45+, F4/80+, Ly6C/G−) microglia. (**C**) LPS treatment (50 ng/mL, 24 h) increases microglial medium L-lactate content, an effect not seen when combined with C43 treatment (100 nM, 1 h post-LPS). (**D**) Neither LPS (50 ng/mL, 24 h) nor C43 (100 nM, 1 h post-LPS) significantly affected primary microglial medium glucose content. Data are mean ± s.e.m. of 3–6 independent cultures.

**Figure 5 ijms-24-15996-f005:**
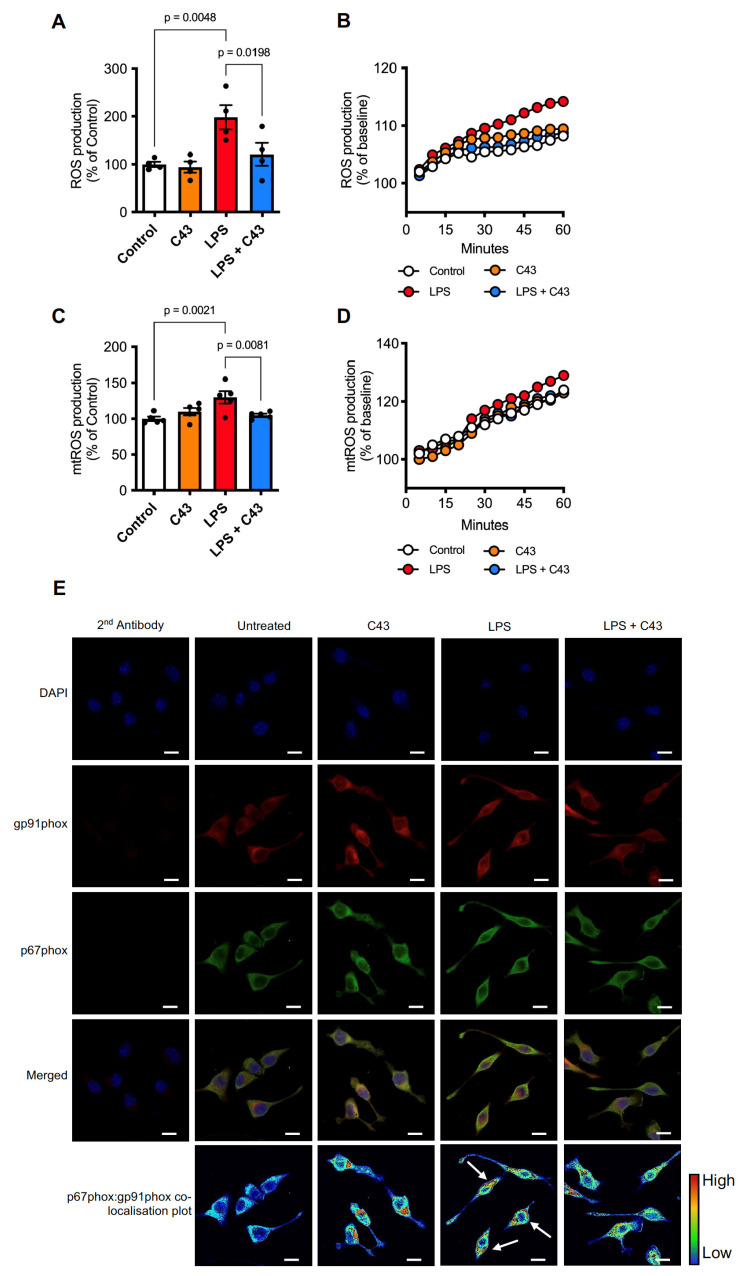
**Fpr2/3 reverses LPS-induced ROS production via effects on mitochondria and NADPH oxidase activity.** (**A**,**B**) Addition of LPS (50 ng/mL) for 1 h increases ROS production in BV2 microglia, an effect completely reversed by subsequent treatment with C43 (100 nM, 10 min post-LPS). (**C**,**D**) Mitochondrial ROS (mtROS) production elicited by LPS exposure (50 ng/mL) for 1 h was also successfully reversed by C43 treatment (100 nM, 10 min post-LPS). (**E**) Treatment of BV2 cells for 30 min with 50 ng/mL LPS stimulated the plasma membrane co-localisation of the NADPH oxidase subunits p67phox (green) and gp91phox (red), an effect reversed by treatment with 100 nM C43 (10 min post-LPS). Nuclei are counterstained with DAPI (blue). Co-localisation intensity of p67phox and gp91phox is represented by the false-colour plots. Graphical data are means ± s.e.m. of 3–5 independent cultures, plated in triplicate. Confocal images are representative of 3 independent cultures; scale bar = 10 μm.

**Figure 6 ijms-24-15996-f006:**
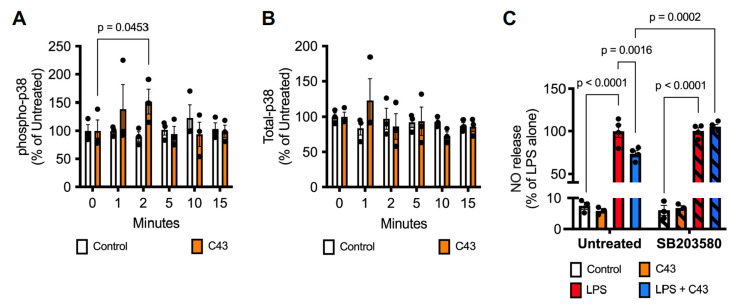
**Activation of Fpr2/3 by C43 stimulates p38 MAPK signalling.** (**A**,**B**) Addition of C43 (100 nM) time-dependently stimulated p38 MAPK phosphorylation, with a maximal response at 2 min post-stimulation. (**C**) Pre-treatment with the selective p38 inhibitor SB203580 (2 µM, 10 min pre-LPS) prevented the reversal of LPS (50 ng/mL, 24 h) induced nitric oxide release effected by C43 treatment (100 nM, 1 h post-LPS). Data are mean ± s.e.m., n = 3–4 independent cultures.

**Figure 7 ijms-24-15996-f007:**
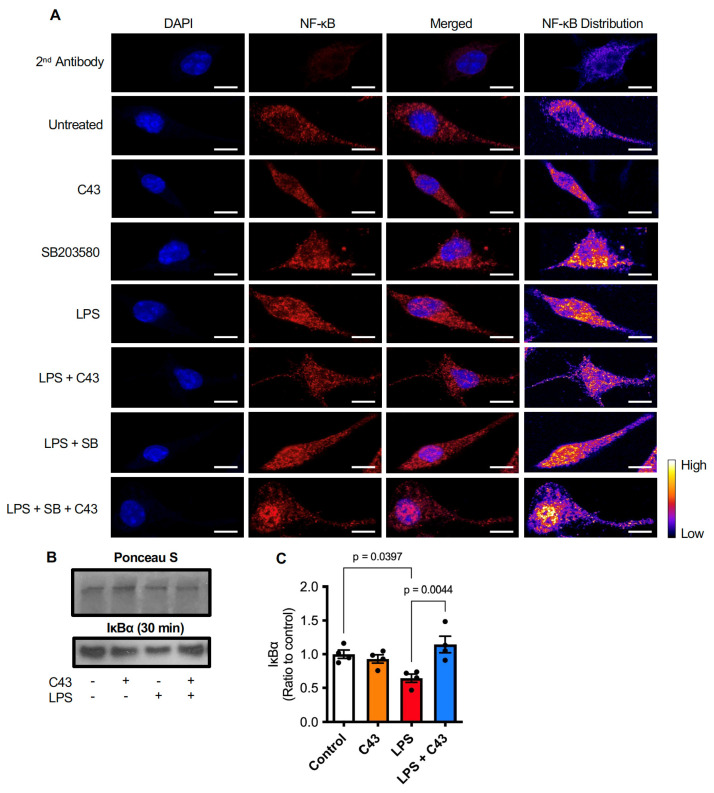
**Fpr2/3 partially reverses LPS-induced NFκB nuclear translocation by preventing IκBα degradation.** (**A**) LPS treatment of BV2 microglia (50 ng/mL, 30 min) stimulates nuclear translocation of the p65 subunit of NFκB (red), an effect partially reversed by treatment with C43 (100 nM, 10 min post-LPS). The effects of C43 are prevented by pre-treatment with the p38 inhibitor SB203580 (2 µM, 5 min prior to C43). Nuclei are counterstained with DAPI, scale bar = 10 µm, images are representative of 3 independent cultures. (**B**,**C**) Treatment of BV2 cells with LPS (50 ng/mL, 30 min) significantly reduced cellular IκBα, an effect prevented by C43 treatment (100 nM, 10 min post-LPS). Graphical data are means ± s.e.m. of 3–4 independent cultures.

## Data Availability

No datasets were generated or analysed during the current study.

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
