# Peer review of "Stimulation of the Pro-Resolving Receptor Fpr2 Reverses Inflammatory Microglial Activity by Suppressing NFκB Activity"

_ijms, 2023, doi:10.3390/ijms242115996_

Round 1
Reviewer 1 Report
Comments and Suggestions for Authors
The authors used murine microglial cell line BV2 and primary microglial cells to investigate the effects of pro-resolving receptor Fpr2 agonist C43 on minimizing LPS-induced microglial responses and activation.
Comments:
1) Errors in lines 186 to 193 – misplaced text of Fig. 4 legend. It should be removed.
2) Errors in line 210 to 223 & 236-237 – it’s the same text as in lines 194-209. It should be removed.
3) Adding C43 after LPS treatment made sense and was a good exp design.
4) It would be more convincing if authors show the Fpr2 expression (and upregulation by LPS) on primary microglial cells and/or BV2 cell lines.
5) BV2 cell lines tend to replicate in cultures. Did LPS treatment (± C43) for 48h have effects on BV2 population?
6) Using primary microglial cultures for all experiments may perhaps provide a more relevant results than using BV2 cell lines since differences can be found between primary cells and cell lines. Here for example, effects of C43 on LPS-induced L-Lactate content in Fig. 1H vs. Fig. 3C were different.
Author Response
Reviewer 1
The authors used murine microglial cell line BV2 and primary microglial cells to investigate the effects of pro-resolving receptor Fpr2 agonist C43 on minimizing LPS-induced microglial responses and activation.
Comments:
1) Errors in lines 186 to 193 – misplaced text of Fig. 4 legend. It should be removed.
We apologise for this error; it has now been corrected.
2) Errors in line 210 to 223 & 236-237 – it’s the same text as in lines 194-209. It should be removed.
We apologise for this error; it has now been corrected.
3) Adding C43 after LPS treatment made sense and was a good exp design.
Thank you for this comment, this was a key part of our strategy, focusing on ways to reverse rather than just prevent neuroinflammation.
4) It would be more convincing if authors show the Fpr2 expression (and upregulation by LPS) on primary microglial cells and/or BV2 cell lines.
We have previously published data showing expression of Fpr2 (and Fpr1) on BV2 cells (McArthur et al. 2010 J Immunol) and have now cited this work as suggested, see lines 78-79.
5) BV2 cell lines tend to replicate in cultures. Did LPS treatment (± C43) for 48h have effects on BV2 population?
We have now included data showing that neither LPS nor C43 treatment, nor their combination, had any effect on BV2 population size following 48 h exposure as suggested, see new Supplemental Figure 3 and lines 99-100.
6) Using primary microglial cultures for all experiments may perhaps provide a more relevant results than using BV2 cell lines since differences can be found between primary cells and cell lines. Here for example, effects of C43 on LPS-induced L-Lactate content in Fig. 1H vs. Fig. 3C were different.
We fully accept that differences can and do occur between immortalised and primary cells in response to inflammatory stimulation, as highlighted in our study of the effects of LPS/C43 on L-lactate production for example. However, we would argue that rather than this being a reason to discard the use of immortalised cells, in truth neither model fully reflects microglial activity in vivo and both are useful tools in establishing the principles of microglial behaviour. In particular, given the inherent inter-batch variability associated with primary microglial cultures, the use of BV2 cells has significant advantages in enabling mechanistic study without killing the substantial numbers of animals that would otherwise be required for primary cell generation.
There may be several potential explanations for the differences we see in the effects of C43 on LPS-induced L-lactate as highlighted by the reviewer, but at least in part these may simply be due to the inherently greater variation in metabolic activity seen between batches of primary microglia when compared to the clonal BV2 cells. Nonetheless, that Fpr2 activation attenuates/reverses inflammatory behaviours in both microglial models, even if the details of this process may vary, underlies the importance of this system in regulating microglial phenotype. It is for this reason that we are careful to state that the work we present is an initial in vitro proof-of-principle and to highlight that future in vivo validation will certainly be necessary. We have now extended the discussion section to emphasise this point, see lines 324-336.
Reviewer 2 Report
Comments and Suggestions for Authors
This study investigates the therapeutic potential of formyl peptide receptor 2 (Fpr2), a key driver of inflammatory resolution, expressed by microglia in neuroinflammation. The researchers explored whether targeting Fpr2 can reverse inflammatory microglial activation triggered by lipopolysaccharide (LPS). The findings reveal that treatment with the Fpr2 agonist C43 significantly attenuates the pro-inflammatory phenotypic changes and reduces the activation of reactive oxygen species (ROS) production in murine primary or immortalized BV2 microglia exposed to LPS. Mechanistic insights indicate that C43 acts through p38 MAPK phosphorylation and prevents LPS-induced NFκB nuclear translocation by inhibiting IκBa degradation. This study emphasizes Fpr2 as a potential target for controlling microglial pro-inflammatory activity, suggesting its potential as a promising therapeutic target for the treatment of neuroinflammatory diseases. This work is well written. I have some comments to improve the manuscript:
1. Thes specificity of agonist (C43) and antagonist (WRW4) used in this study: FPR1, 2 and 3 are similar proteins and these proteins are enriched in microglia. FPR1 has 69% amino acid identity with FPR2 and 56% with FPR3, whereas FPR2 and FPR3 share 83% identity. The small molecules are usually not just selectively targeting one of the FPRs. The difference is affinity. For example, C43 is the agonist of FPR1 and FPR2, and WRW4 is an antagonist of FPR2 and FPR3. Therefore, if the authors claim that the C43 inhibits microglia mediated neuroinflammation via FPR2, the results from pharmacological experiments are not enough. I would expect to see the following results by manipulating microglial FPRs:
a. knockdown/knockout Fpr2 attanuates the effect of C43 on neuroinflammation.
b. knowckdown/knockout Fpr1 does not change the effect of C43 on neuroinflammation.
Otherwise authors may change their conclusions and by replacing Fpr2 with Fprs.
2. Lines 93-94: but when given 1 h after administration of LPS (50 ng/ml) C43 significantly increased bacterial uptake approximately 8-fold after 24 h (Figure 1G). This sentence is confusing, and according to Figure 1G and the corresponding legend, the phagocytosis is attenuated by C43.
Author Response
This study investigates the therapeutic potential of formyl peptide receptor 2 (Fpr2), a key driver of inflammatory resolution, expressed by microglia in neuroinflammation. The researchers explored whether targeting Fpr2 can reverse inflammatory microglial activation triggered by lipopolysaccharide (LPS). The findings reveal that treatment with the Fpr2 agonist C43 significantly attenuates the pro-inflammatory phenotypic changes and reduces the activation of reactive oxygen species (ROS) production in murine primary or immortalized BV2 microglia exposed to LPS. Mechanistic insights indicate that C43 acts through p38 MAPK phosphorylation and prevents LPS-induced NFκB nuclear translocation by inhibiting IκBa degradation. This study emphasizes Fpr2 as a potential target for controlling microglial pro-inflammatory activity, suggesting its potential as a promising therapeutic target for the treatment of neuroinflammatory diseases. This work is well written. I have some comments to improve the manuscript:
- Thes specificity of agonist (C43) and antagonist (WRW4) used in this study: FPR1, 2 and 3 are similar proteins and these proteins are enriched in microglia. FPR1 has 69% amino acid identity with FPR2 and 56% with FPR3, whereas FPR2 and FPR3 share 83% identity. The small molecules are usually not just selectively targeting one of the FPRs. The difference is affinity. For example, C43 is the agonist of FPR1 and FPR2, and WRW4 is an antagonist of FPR2 and FPR3. Therefore, if the authors claim that the C43 inhibits microglia mediated neuroinflammation via FPR2, the results from pharmacological experiments are not enough. I would expect to see the following results by manipulating microglial FPRs:
- knockdown/knockout Fpr2 attanuates the effect of C43 on neuroinflammation.
- knowckdown/knockout Fpr1 does not change the effect of C43 on neuroinflammation.
Otherwise authors may change their conclusions and by replacing Fpr2 with Fprs.
We appreciate the reviewer’s point and have now included additional data with the specific FPR1 antagonist cyclosporin H, a drug with essentially no binding activity at FPR2 (Stenfeldt et al. 2007 Inflammation). In our hands, pre-treatment with cyclosporin H had no effect on the ability of C43 to ameliorate LPS-induced production of either nitric oxide or TNFα production, in marked contrast to the effects of WRW4 (see new Figure 2D-E and lines 121-134). We are therefore confident the effects of C43 are unlikely to be mediated through FPR1.
As we acknowledge in the introduction, the murine receptors Fpr2 and Fpr3 are close functional analogues, essentially mirroring the actions of the human FPR2. There are currently no agents reliably able to distinguish between murine Fpr2 and Fpr3, hence we have rephrased Fpr2 as Fpr2/3 as suggested by the reviewer.
- Lines 93-94: but when given 1 h after administration of LPS (50 ng/ml) C43 significantly increased bacterial uptake approximately 8-fold after 24 h (Figure 1G). This sentence is confusing, and according to Figure 1G and the corresponding legend, the phagocytosis is attenuated by C43.
We apologise for this error and have rewritten this section accordingly to ensure clarity, see lines 88-96.
Round 2
Reviewer 2 Report
Comments and Suggestions for Authors
My comments were carefully addressed. I do not have further concerns.